# Inhibition of Rice Stripe Virus Accumulation by Polyubiquitin-C in *Laodelphax striatellus*

**DOI:** 10.3390/insects15030149

**Published:** 2024-02-22

**Authors:** Bo-Xue Li, Yu-Hua Qi, Chuan-Xi Zhang, Jian-Ping Chen, Jun-Min Li, Gang Lu

**Affiliations:** State Key Laboratory for Managing Biotic and Chemical Threats to the Quality and Safety of Agro-Products, Key Laboratory of Biotechnology in Plant Protection of MARA and Zhejiang Province, Institute of Plant Virology, Ningbo University, Ningbo 315211, China

**Keywords:** rice stripe virus, polyubiquitin-C, *Laodelphax striatellus*, viral accumulation

## Abstract

**Simple Summary:**

The ubiquitin system plays an important role in defense against viruses. In this study, we cloned and characterized the polyubiquitin-C (UBC) gene in *Laodelphax striatellus*. Expression profile analysis showed that *LsUBC* was highly expressed in the salivary glands, midgut, and reproductive system. Moreover, the expression of *LsUBC* was upregulated during viral infection, and inhibition of *LsUBC* expression increased viral accumulation. These results indicated that *LsUBC* plays a vital role in inhibiting viral accumulation in *L. striatellus*.

**Abstract:**

Many hosts utilize the ubiquitin system to defend against viral infection. As a key subunit of the ubiquitin system, the role of polyubiquitin in the viral infection of insects is unclear. Here, we identified the full-length cDNA of the polyubiquitin-C (UBC) gene in *Laodelphax striatellus*, the small brown planthopper (SBPH). *LsUBC* was expressed in various tissues and was highly expressed in salivary glands, midgut, and reproductive systems. Furthermore, the *LsUBC* expression profiles in the developmental stages showed that *LsUBC* was ubiquitously expressed in seven developmental stages and was highest expressed in female adults with SBPH. qRT-PCR analyses indicated that rice stripe virus (RSV) infection promoted the *LsUBC* expression. Knockdown of *LsUBC* mRNA via RNA interference increased RSV accumulation. These findings suggest that LsUBC inhibits RSV accumulation in *L. striatellus*.

## 1. Introduction

Ubiquitin is a family of highly conserved proteins in eukaryotic organisms and is involved in the regulation of many cellular events [1,2]. In the canonical function of ubiquitin, the C-terminal glycine residue of ubiquitin can be linked to the epsilon-amino group of a lysine residue in substrate proteins through the action of ubiquitin-protein ligase [3,4]. Genes encoding ubiquitin can be divided into monomeric and polymeric ubiquitin genes. The monomeric can be fused to other carboxyl-extension protein-coding regions. The polyubiquitin genes contain multiple tandem ubiquitin repeat regions with no spacer sequences between monomer units [5]. Polyubiquitin-C (UBC) is one of the sources of ubiquitin and is a stress-regulated polyubiquitin gene. UBC proteins play an essential role in maintaining ubiquitin levels and cellular homeostasis under stress conditions in mammals [6,7]. Unanchored ubiquitin chains are reported to connect the proteasome and autophagy to eliminate abnormal or foreign protein accumulation [8].

Numerous hosts (e.g., plants, insects, and mammals) utilize the ubiquitin system to defend against viral infection. For example, ubiquitin-like protein 5 directly interacted with rice stripe virus (RSV) P3 protein to mediate the viral P3 degradation through the 26S proteasome system in plants [9]. Also, ubiquitin-proteasome system could decrease the replication of the Turnip yellow mosaic virus by regulating the accumulation of viral protein [10]. Conversely, viruses have evolved many strategies to exploit the ubiquitin system to evade host immune responses. For example, Tomato yellow leaf curl virus C2 protein interacted with ubiquitin-40S ribosomal protein S27a (RPS27A) to compromise the degradation of JAZ1 protein and inhibit plant jasmonic acid defense [11]. In addition, both rice black-streaked dwarf virus (RBSDV) and RSV interfered with insect 26S proteasome to attenuate plant defense response and facilitated virus accumulation and transmission [12,13]. Similarly, human immunodeficiency virus type 1 (HIV-1) hijacked ubiquitin systems to evade host interferon responses in mammals [14].

The small brown planthopper (SBPH, *Laodelphax striatellus*) is one of the most serious pests and has caused severe losses in rice production in East Asia. SBPH also acts as a vector for transmission of rice stripe virus (RSV) in a persistent and circulative-propagative manner [15,16,17]. RSV is a typical member of the genus *Tenuivirus* and contains four single-stranded, negative-sense RNA segments that encode seven viral proteins. RSV particles first establish infection in the midgut epithelium, then spread to various tissues and ultimately reach the salivary glands or ovaries under the transportation of hemolymph [15,18]. Recently, several studies have reported that multiple proteins in the ubiquitin system are involved in viral infection and inhibit virus accumulation in SBPHs [12,13,19]. However, the ubiquitin UBC protein involved in RSV infection has not been identified to date.

In this study, we cloned and characterized the *L. striatellus UBC* (*LsUBC*). The expression of *LsUBC* was upregulated during viral infection. In addition, inhibition of *LsUBC* expression facilitated RSV accumulation in *L. striatellus*. These results indicated that *LsUBC* plays a negative role in mediating viral infection.

## 2. Materials and Methods

### 2.1. Virus, Plants and Insects

The RSV-infected rice plants were originally obtained from Jiangsu Province, China, and were maintained in the laboratory. The viruliferous (RSV-infected) and nonviruliferous (RSV-free) SBPHs were reared separately on rice plants inside a growth incubator at 26 °C with 80% relative humidity and a photoperiod of 14 h light and 10 h dark. To ensure a high infection rate, the offspring of viruliferous females were monitored and collected for RT-PCR assay. The infection rate of the viruliferous population used in the experiments was approximately 90%.

### 2.2. Identification and Phylogenetic Analysis of LsUBC

To identify the putative UBC sequence in *L. striatellus*, annotated UBC sequences in other insect species were collected and used as a query to compare similar sequences in the SBPH genome [20]. The candidate sequences with high E-values were extracted and confirmed by Blastp search against a non-redundant protein database at the National Center for Biotechnology Information (NCBI). The open reading frame sequences were recovered with the ORF Finder program (https://www.ncbi.nlm.nih.gov/orffinder; accessed on 27 June 2023). The sequence function annotation was analyzed using the NCBI Conserved Domain Database (https://www.ncbi.nlm.nih.gov/cdd/; accessed on 27 June 2023). A phylogenetic tree was obtained by the ML algorithm using RAxML-NG with 1000 bootstrap replications [21].

### 2.3. Tissue Collection

Adult SBPHs were collected and dissected using tweezers. Each type of tissue sample was obtained from 20 female or male insects. The dissected tissues were washed twice with 1xPBS and stored with the RNAiso Plus solution (Takara, Beijing, China) at –80 °C. Six tissue types were isolated (epidermis, salivary glands, midgut, ovary, testis, and fat body) and three replicates per tissue were collected for expression analysis of the *LsUBC* gene.

### 2.4. RNA Extraction and qRT-PCR

Total RNA was extracted from whole insects or tissue samples using RNAiso Plus solution. Total RNA was then transcribed into complementary DNA (cDNA) using the HiScript II Q RT SuperMix for qPCR (+gDNA wiper) (Vazyme, China) following the standard protocols. Quantitative real-time PCR (qRT-PCR) was performed using Hieff qPCR SYBR Green Master Mix (Yeasen, China). The expression level of the *LsUBC* gene was calculated with the 2^−ΔΔCt^ method and a housekeeping gene (*Actin*) was amplified as an internal standard. Three biological replicates were carried out for each experiment, and two technical replicates were conducted for each biological replicate.

### 2.5. Western Blotting

SBPH samples were collected and ground with liquid nitrogen. Total protein samples were then collected from the extraction buffer and treated with 6× SDS loading buffer. After boiled for 10 min, the samples were separated by 10% SDS-PAGE gels and transferred onto polyvinylidene fluoride (PVDF) membranes. The membranes were inoculated with primary antibodies against RSV-NP (1:5000 dilution) or ACTB (1:5000 dilution) followed by probed with horseradish peroxidase (HRP)-conjugated goat anti-mouse antibody at 1:10,000 dilution. The membranes were detected using the Luminescent Image Analyzer AI680 (GE, Danderyd, Sweden).

### 2.6. RNA Interference (RNAi)

The dsRNAs targeting *LsUBC* and *GFP* genes were synthesized using the T7 High Yield Transcription Kit as recommended by the manufacturer (Vazyme, Nanjing, China). The PCR primers with T7 promoter sequences are available in Appendix A. Third-instar nymphs were microinjected with 25 nl of ds*LsUBC* or ds*GFP* into insect hemolymph using a TransferMan 4r micromanipulator (Eppendorf, Hamburg, Germany). Following microinjection, nymphs were transferred to healthy rice seedlings until used for qRT-PCR or Western blotting analysis.

## 3. Results

### 3.1. Identification and Characterization of LsUBC

The sequence of the *LsUBC* gene was obtained by comparing the SBPH genome with *UBC* genes from other insect species. This sequence was then compared to the NCBI non-redundant protein database through Blastp search. The result showed that the amino acid sequence of LsUBC was highly conserved with the UBC sequences of other species and shared the highest identity (90%) with the *Pimephales promelas* UBC. Subsequently, the *LsUBC* gene was confirmed by PCR amplification and verified by Sanger sequencing.

The ORF sequence of *LsUBC* (GenBank accession number OR944057.1) was 1137 bp in length and encoded 378 amino acids with a calculated molecular mass of 42 kDa. Conserved domain analysis by NCBI Conserved Domain Database search found that *LsUBC* contained five repeated ubiquitin homologues (UBQ) (Figure 1A). Phylogenetic analysis using amino acid sequences showed that LsUBC clustered with other insect UBCs deposited in the NCBI database with a high branch support (Figure 1B).

Sequence alignment analysis using UBCs from *L. striatellus* and other insects revealed that these UBCs were highly conserved in each insect species and contained multiple tandem repeating monomeric units (76 amino acids). In addition, several amino acids at the C-terminus of LsUBC are not conserved (Figure 2).

### 3.2. Expression Analysis of LsUBC in SBPHs

To investigate the detailed expression profiles, we performed a qRT-PCR assay to examine the expression of *LsUBC* in different tissues and developmental stages of SBPH. The results in Figure 3A showed that *LsUBC* was expressed in all six tissues. Meanwhile, *LsUBC* was mainly highly expressed in four tissues (salivary glands, midgut, ovaries, and testis) and most highly expressed in ovaries (Figure 3A). In addition, the *LsUBC* expression profiles in the seven developmental stages were also detected. The results showed that *LsUBC* was ubiquitously expressed in seven collected developmental stages of SBPH. The highest transcription level of *LsUBC* was observed in female adult samples (Figure 3B).

### 3.3. Rice Stripe Virus Infection Upregulates the Expression of LsUBC in SBPHs

To determine whether the *LsUBC* was involved in viral infection, we compared the expression of *LsUBC* between RSV-infected and RSV-free SBPH adults. Compared with the RSV-free SBPHs, the transcript level of the *LsUBC* was significantly increased in the RSV-infected insect population (Figure 4A). Moreover, we conducted a time course study to detect the expression of Ls*UBC* in SBPHs. After 3 days of feeding on RSV-infected rice plants, SBPH samples were transferred to un-infected rice seedlings and collected at 2, 4, and 6 days post RSV feeding. To eliminate the effect of the insect developmental stage on *LsUBC*, another group of SBPHs feeding on rice without RSV was collected as a control. Compared with the controls, the transcript level of the *LsUBC* was higher in RSV-acquired SBPH groups (Figure 4B). These results indicated that RSV infection increases *LsUBC* expression in SBPHs.

### 3.4. Inhibition of LsUBC Expression Increases RSV Accumulation in SBPHs

To further explore the effect of LsUBC on RSV accumulation, RSV-infected third-instar nymphs were microinjected with dsRNAs targeting *LsUBC* (ds*LsUBC*) or *GFP* (ds*GFP*). At two days post microinjection, we first examined the expression of *LsUBC*. Compared with the ds*GFP*-treated group, the transcript level of *LsUBC* was significantly lower in the ds*LsUBC*-treated group. Further, we examined the accumulation of RSV nucleocapsid protein (NP) at the RNA level and protein level. As shown in Figure 5B, the RNA level of RSV NP was significantly increased in the ds*LsUBC*-treated SBPHs compared with the ds*GFP* controls. Similarly, the amount of RSV NP in ds*LsUBC*-treated SBPHs also obviously increased, as determined by Western blotting assay (Figure 5C). These results suggested that inhibition of *LsUBC* expression facilitated RSV accumulation in SBPHs.

## 4. Discussion

As a persistently transmitted virus, RSV needs to accumulate in the insect vector. At the same time, numerous antiviral immune systems in insects are activated to inhibit excessive RSV accumulation. Among them, the ubiquitin system plays an important role in resisting viral infection and spread in SBPHs. Here, we characterized a polyubiquitin gene *LsUBC* and explored its effect on viral infection. The results indicate that *LsUBC* plays a positive role in inhibiting excessive accumulation of RSV NP in *L. striatellus*.

Sequence alignment of LsUBC and other insect UBCs revealed these sequences all have highly tandem repeating monomeric units. However, the number of monomeric units varies considerably among insect species. The polyubiquitin gene family was generally thought to be subjected to concerted evolution. Currently, there are two mechanisms for operating the evolution of the UBC gene, namely gene conversion and unequal crossing-over [22,23,24,25]. In *C. elegans*, and *A. thaliana*, the monomeric units within the polyubiquitin gene were highly diverged in silent substitution sites. In the evolution of the mouse UBC gene, unequal crossing-over was considered to be the main mechanism of homogenization within the rodent lineage [25]. Whether there is a similar mechanism for the evolution of insect UBC warrants further investigation.

Although expressed in all SBPH tissues, *LsUBC* is highly expressed in salivary glands, the midgut, and the reproductive system. This tissue distribution is closely related to the process of RSV infection. During the long-term co-evolution of insects and viruses, RSV infection activates the ubiquitin system and protects against RSV accumulation. Other subunits in the ubiquitin system have been reported to be involved in virus accumulation in insect vectors. For example, the silencing of two regulatory particles non-ATPase (RPN3 and RPN8) increased RSV accumulation and viral proteins could hijack the two RPN proteins to facilitate virus accumulation [12,13]. In *Culex* mosquitoes, West Nile virus infection increased the expression of cullin RING ubiquitin ligase Cullin4. Knockdown of Cullin4 activated the JAK-STAT pathway and inhibited viral replication [26]. Similarly, another Cullin3 protein has also been reported to be a proviral host factor during chikungunya virus infection in *Aedes aegypti* [27]. We show that the expression of *LsUBC* increased during RSV infection and repression of *LsUBC* facilitated RSV accumulation. The molecular mechanism of how LsUBC protein inhibits RSV accumulation remains to be further explored.

In conclusion, the full-length cDNA of *LsUBC* was obtained and sequenced in *L. striatellus*. The expression of *LsUBC* was highly expressed in salivary glands, midgut, and reproductive systems. In addition, *LsUBC* was highly expressed in RSV-infected SBPHs and inhibited RSV accumulation in *L. striatellus*. These results indicated that the expression of *LsUBC* correlated with RSV accumulation in SBPHs. This study provides new insight into the persistent RSV infection in SBPH.

## Figures and Tables

**Figure 1 insects-15-00149-f001:**
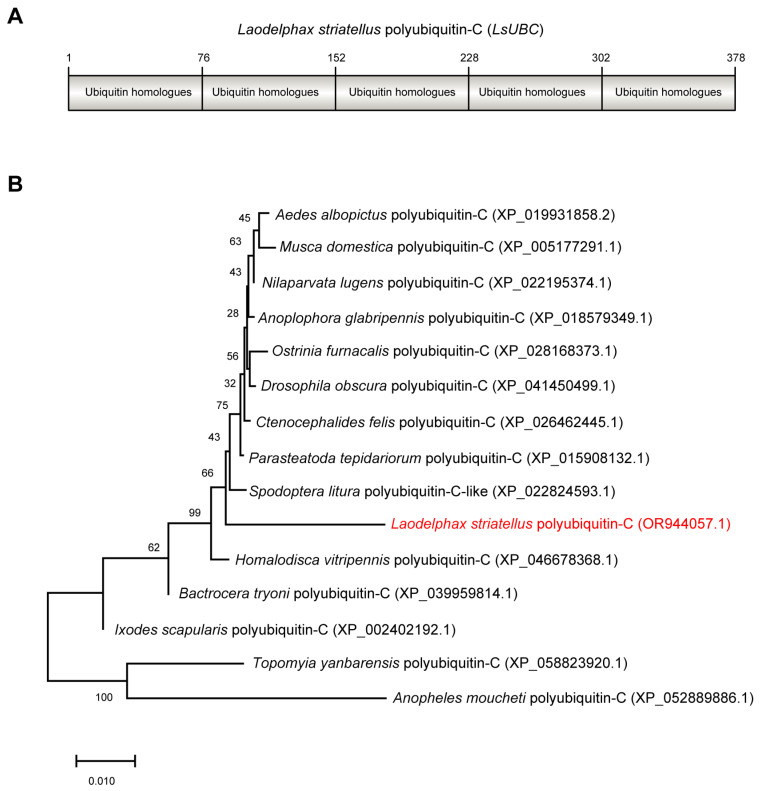
Sequence characterization and phylogenetic analysis of *Laodelphax striatellus* polyubiquitin-C (*LsUBC*) gene. (**A**) Schematic diagrams showing the amino acid sequence of *LsUBC*. The position of each single monomeric unit is marked. (**B**) Phylogenetic analysis using the amino acid sequences of polyubiquitin-C from *L. striatellus* and other insect species. A phylogenetic tree was obtained by the ML algorithm using RAxML-NG with 1000 bootstrap replications. The position of the *LsUBC* on the phylogenetic tree is labeled in red font.

**Figure 2 insects-15-00149-f002:**
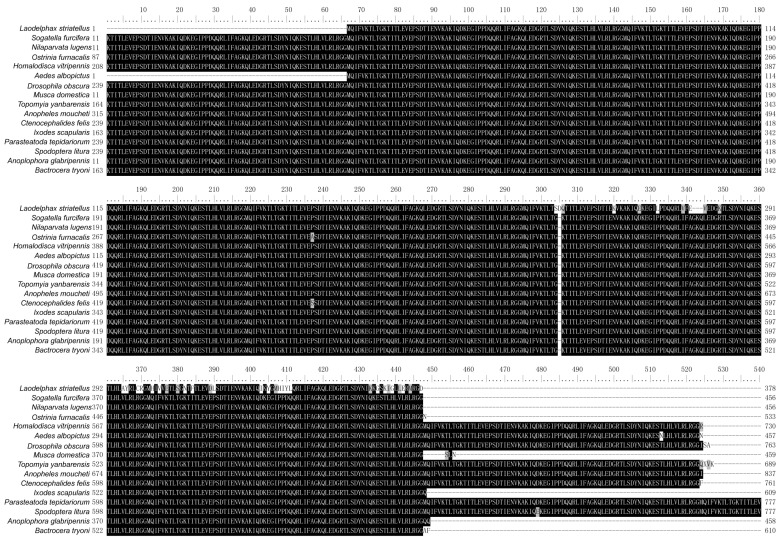
Sequence alignment of polyubiquitin-C from *L. striatellus* and other insect species. The amino acid sequences of UBC from the following 16 insect species were aligned: *L. striatellus*; *Sogatella furcifera*; *Nilaparvata lugens*; *Ostrinia furnacalis*; *Homalodisca vitripennis*; *Aedes albopictus*; *Drosophila obscura*; *Musca domestica*; *Topomyia yanbarensis*; *Anopheles moucheti*; *Ctenocephalides felis*; *Ixodes scapularis*; *Parasteatoda tepidariorum*; *Spodoptera litura*; *Anoplophora glabripennis*; *Bactrocera tryoni*. Residues that are fully conserved are shaded in black, and those that are partially conserved are shown in gray.

**Figure 3 insects-15-00149-f003:**
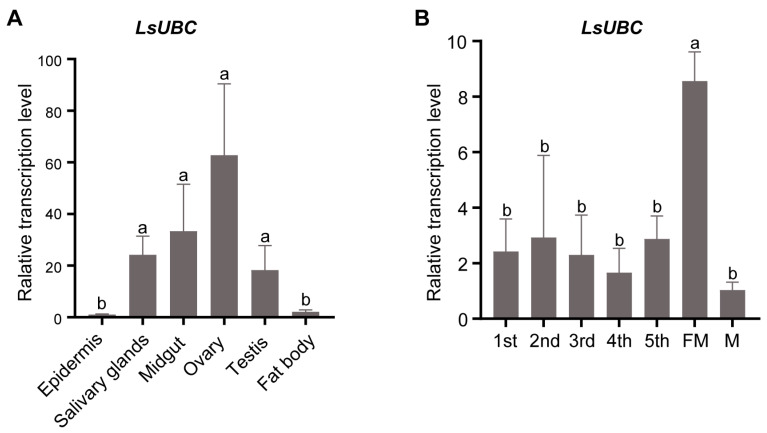
Analyses of *LsUBC* expression in different tissues and different developmental stages of *L. striatellus*. (**A**) qRT-PCR analysis of *LsUBC* expression in six different tissues. Similar *LsUBC* expression levels were found in the epidermis and fat body, whereas high expression was detected in the salivary glands, midgut, ovaries, and testis. Tissues obtained from 50 RSV-free SBPH adults were considered to be a single replicate, and the experiment was performed in three independent replicates. (**B**) qRT-PCR analysis of *LsUBC* expression in seven developmental stages. A total of 15 RSV-free SBPHs were considered to be a single replicate, and the experiment contained three replicates. Different letters indicate significant differences in *LsUBC* expression levels (*p* < 0.05).

**Figure 4 insects-15-00149-f004:**
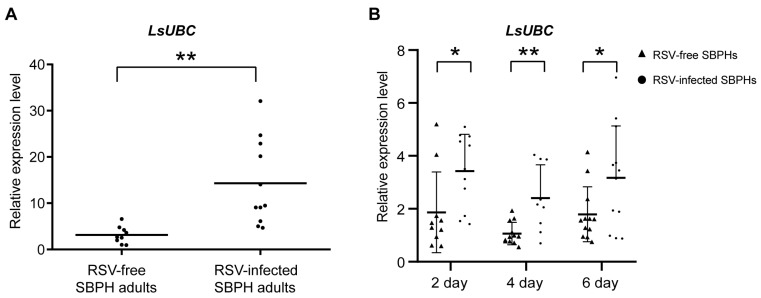
Analyses of the *LsUBC* expression during RSV infection in *L. striatellus*. (**A**) qRT-PCR analysis of *LsUBC* expression in RSV-free and RSV-infected SBPH adults. (**B**) qRT-PCR analysis of *LsUBC* expression at 2, 4, and 6 days post RSV feeding, respectively. SBPHs feeding on rice without RSV were considered as a control (RSV-free SBPHs). Each dot represents a single insect sample. Experiments were performed on three independent replicates. *, *p* < 0.05; **, *p* < 0.01.

**Figure 5 insects-15-00149-f005:**
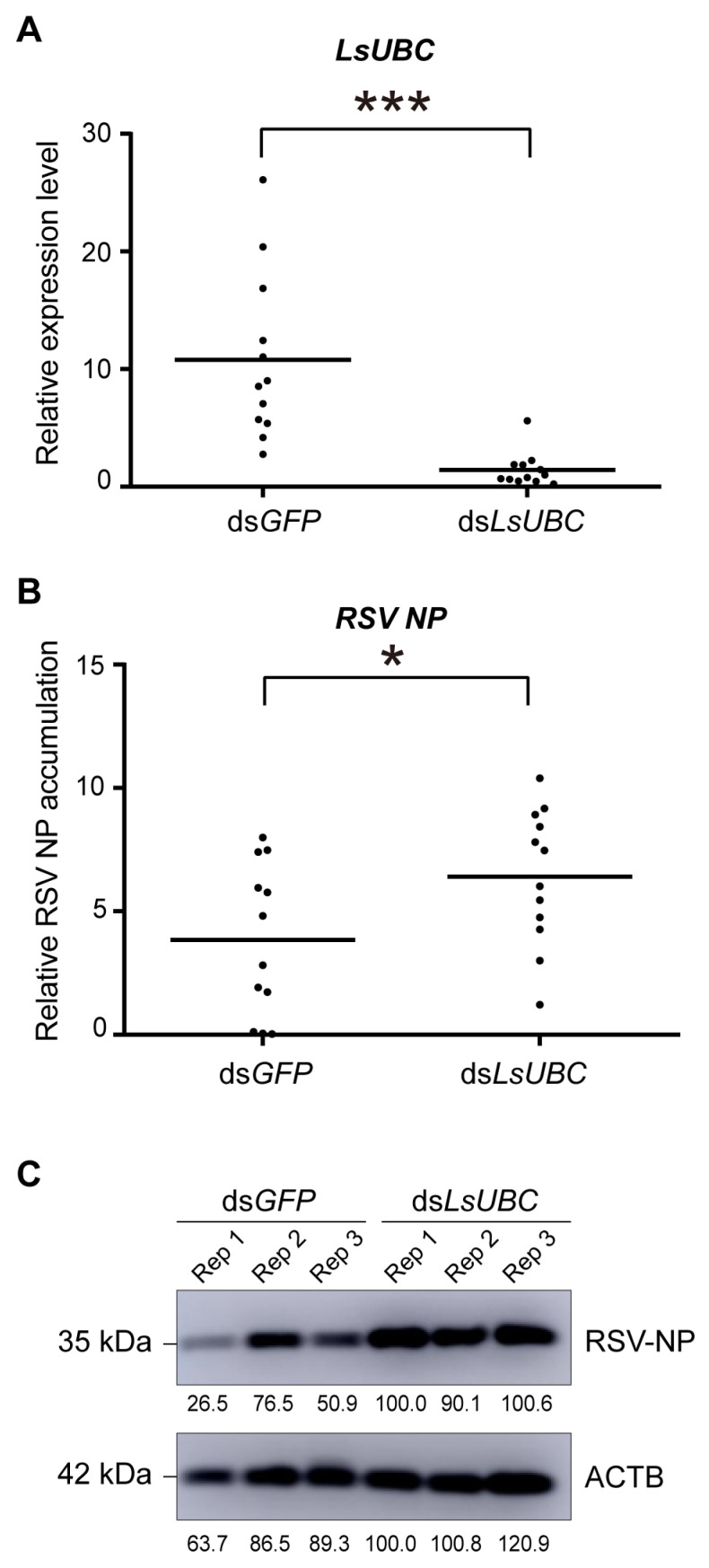
Effect of *LsUBC* knockdown on RSV accumulation in *L. striatellus*. The expression of *LsUBC* (**A**) and RSV nucleocapsid protein (NP) (**B**) in dsGFP- and dsLsUBC-treated RSV-infected SBPHs were determined by qRT-PCR. Each dot represents a single insect sample. Experiments were performed on three independent replicates. *, *p* < 0.05; ***, *p* < 0.001. (**C**) The protein level accumulation of RSV NP in dsGFP- and dsLsUBC-treated RSV-infected SBPHs was detected using a Western blotting assay. ACTB was used as a loading control.

## Data Availability

The data presented in this study are openly available in GenBank (OR944057.1).

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
