# Peer review of "Inhibition of Rice Stripe Virus Accumulation by Polyubiquitin-C in *Laodelphax striatellus"

_insects, 2024, doi:10.3390/insects15030149_

Round 1
Reviewer 1 Report
Comments and Suggestions for Authors
The authors cloned a UBC gene from SBPH and conducted sequence analysis. Next, the authors analyzed its expression in RSV-infected and un-infected SBPH. Finally, the authors used RNAi to explore the role of UBC in SBPH resistance against RSV. In general, this study is well designed and the results obtained are informative. The findings contribute to a better understanding of SBPH-RSV interactions, and shed light on insect vector-plant virus interactions in general. I would suggest acceptance after the authors resolve the following minor issues.
Line 14 and below how do you get the conclusion ‘inhibiting excessive viral accumulation’. I would like to suggest the deletion of excessive as you have no proof of this.
Line 48 and 65, and below change besides to in addition
Line 54 Asian should be Asia
Line 88 add some details about dissection, for example what tools you used and how you dissect. And add details about the insects you dissect, for example age and gender.
Line 139 LsUBC italic
Line 169 change promotes to upregulates
Line 175 change healthy to un-infected
Line 204 delete viral load
Discussion section the second and third paragraph should switch. You should first talk about UBC sequences and then its expression and function.
Comments on the Quality of English LanguageThere are some minor issues in writing. Please go through the manuscript carefully to resolve these issues.
Author Response
Response to Reviewers:
Reviewer #1: The authors cloned a UBC gene from SBPH and conducted sequence analysis. Next, the authors analyzed its expression in RSV-infected and un-infected SBPH. Finally, the authors used RNAi to explore the role of UBC in SBPH resistance against RSV. In general, this study is well designed and the results obtained are informative. The findings contribute to a better understanding of SBPH-RSV interactions, and shed light on insect vector-plant virus interactions in general. I would suggest acceptance after the authors resolve the following minor issues.
Response: We thank the reviewer #1 for his/her great comments and suggestions. We have revised the manuscript accordingly.
Line 14 and below how do you get the conclusion ‘inhibiting excessive viral accumulation’. I would like to suggest the deletion of excessive as you have no proof of this.
Response: We have deleted it (Line 14).
Line 48 and 65, and below change besides to in addition
Response: We have modified these sentences (Lines 48, 65 142 and 239).
Line 54 Asian should be Asia
Response: We have revised it (Line 14).
Line 88 add some details about dissection, for example what tools you used and how you dissect. And add details about the insects you dissect, for example age and gender.
Response: Thanks for your good suggestions. We have now modified the text (Lines 88-89).
Line 139 LsUBC italic
Response: We have done it (Line 139).
Line 169 change promotes to upregulates
Response: We have changed it (Line 169).
Line 175 change healthy to un-infected
Response: We have changed it (Line 175).
Line 204 delete viral load
Response: We have deleted it (Line 205).
Discussion section the second and third paragraph should switch. You should first talk about UBC sequences and then its expression and function.
Response: Thank you for this suggestion. We have changed the two paragraphs (Lines 211-236).
There are some minor issues in writing. Please go through the manuscript carefully to resolve these issues.
Response: Thanks for your good suggestions. We have modified the language of the manuscript to make it suitable for publication.
Reviewer 2 Report
Comments and Suggestions for Authors
The paper entitled “Inhibition of Rice Stripe Virus Accumulation by Polyubiquitin-C in Laodelphax striatellus” by Li et al reports that LsUBC was highly expressed in salivary glands, midgut and reproductive system and upregulated during viral infection. Moreover, inhibition of LsUBC expression increased viral accumulation.
Major concerns:
Ubiquitin is encoded by different polyubiquitin genes in different species. For example, in human, there are four ubiquitin genes, UBB, RPS27A, UBA52, and UBC, as well as at least 52 pseudogenes of these genes. Please explain why LsUBC was selected from ployubiquitin genes for research?
A lot of evidence has shown that polyubiquitination has a "double-edged sword" effect in RSV virus infection. I suggest trying to supplement the discussion on how LsUBC protects against RSV accumulation.
Figure 1. Why the homology of polyubiquitin-C in the phylogenetic tree is more closely related to Lepidoptera (Spodoptera litura), while showing lower homology with Nilaparvata lugens? The same as sequence alignment in Figure 2. Please provide an explanation.
Minor concerns:
Figure 3. Please explain why the level of LsUBC in adult females is higher than in larvae and adult males? The specific expression indicates that LsUBC has special functions in adult females. Are the samples of females and males in a 1:1 ratio in the subsequent experiments?
Figure 5B. Why do RSV NP levels have such high variability? I suggest increasing the number of replicates, despite there is significant difference between dsUBC-treated group and control.
Figure 5C. It is inappropriate to use ACTB as a loading control. LsActin has been reported to play a role in RSV transmission. I suggest changing the loading control.
Line 38-39, please check the references 8 here, because the conclusions cited here do not exist in this reference. “Unanchored UBC” should be changed to “unanchored ubiquitin chains”. The study of references 8 reported that unanchored ubiquitin chains connect and coordinate the proteasome and autophagy to eliminate toxic or foreign proteins aggregate.
Line 128, “cDNA” should be changed to “ORF” or “CDS”, because the sequences of LsUBC (GenBank accession number OR944057.1) lack 3’UTR and 5’UTR.
Line 213-215, I suggest deleting the sentence about RSV Infection route in SBPH, as this information is presented in introduction section at lines 57-60.
The band intensity of western blots should be quantified.
Please checked the original data of Figure 5B. The amount of sample in dsGFP and dsUBC are inconsistent.
Figure 5A and 5B probably come from the same experiment. The amount of sample in Figure 5A and Figure 5B are inconsistent.
Author Response
Response to Reviewers:
Reviewer #2: The paper entitled “Inhibition of Rice Stripe Virus Accumulation by Polyubiquitin-C in Laodelphax striatellus” by Li et al reports that LsUBC was highly expressed in salivary glands, midgut and reproductive system and upregulated during viral infection. Moreover, inhibition of LsUBC expression increased viral accumulation.
Response: We thank reviewer#2 for his/her comments. We have revised the manuscript accordingly.
Major concerns:
Ubiquitin is encoded by different polyubiquitin genes in different species. For example, in human, there are four ubiquitin genes, UBB, RPS27A, UBA52, and UBC, as well as at least 52 pseudogenes of these genes. Please explain why LsUBC was selected from ployubiquitin genes for research?
Response: We are sorry for this confusion. Although polyubiquitin genes have been reported to be involved in viral infection, the function of UBC in plant virus infection is still unknown. Our results showed that LsUBC could inhibit RSV accumulation in L. striatellus and this gene may have similar function in other insects.
A lot of evidence has shown that polyubiquitination has a "double-edged sword" effect in RSV virus infection. I suggest trying to supplement the discussion on how LsUBC protects against RSV accumulation.
Response: Thanks for your good suggestions. We have tried to find the relationship between LsUBC and RSV through protein-protein interactions, but we have not been able to find viral proteins that could interact with LsUBC. The molecular mechanism of how LsUBC protein inhibits RSV accumulation remains to be further explored.
Figure 1. Why the homology of polyubiquitin-C in the phylogenetic tree is more closely related to Lepidoptera (Spodoptera litura), while showing lower homology with Nilaparvata lugens? The same as sequence alignment in Figure 2. Please provide an explanation.
Response: Polyubiquitin-C genes are highly conserved in multiple insect species. Therefore, LsUBC is shown to be highly related to other species. Figure 1B and Figure 2 only showed that LsUBC clusters with other insect UBCs.
Minor concerns:
Figure 3. Please explain why the level of LsUBC in adult females is higher than in larvae and adult males? The specific expression indicates that LsUBC has special functions in adult females. Are the samples of females and males in a 1:1 ratio in the subsequent experiments?
Response: Thanks for your critical comments. We also do not know why the level of LsUBC is higher in adult females. We mixed females and males out of proportion in the experiments.
Figure 5B. Why do RSV NP levels have such high variability? I suggest increasing the number of replicates, despite there is significant difference between dsUBC-treated group and control.
Response: We speculate that the differences in insect immunity during RSV infection contribute to the high variability in RSV NP levels. We have increased the number of replicates (Figure 5B).
Figure 5C. It is inappropriate to use ACTB as a loading control. LsActin has been reported to play a role in RSV transmission. I suggest changing the loading control.
Response: Although ACTB has been reported to be involved in RSV transmission, dsLsUBC or dsGFP was injected into the same background (RSV-infected nymphs).
Line 38-39, please check the references 8 here, because the conclusions cited here do not exist in this reference. “Unanchored UBC” should be changed to “unanchored ubiquitin chains”. The study of references 8 reported that unanchored ubiquitin chains connect and coordinate the proteasome and autophagy to eliminate toxic or foreign proteins aggregate.
Response: Thanks for your good suggestions. We have revised it (Line 38).
Line 128, “cDNA” should be changed to “ORF” or “CDS”, because the sequences of LsUBC (GenBank accession number OR944057.1) lack 3’UTR and 5’UTR.
Response: We have changed it (Line 128).
Line 213-215, I suggest deleting the sentence about RSV Infection route in SBPH, as this information is presented in introduction section at lines 57-60.
Response: We have deleted this sentence.
The band intensity of western blots should be quantified.
Response: We have modified it (Figure 5C).
Please checked the original data of Figure 5B. The amount of sample in dsGFP and dsUBC are inconsistent.
Response: Thanks for point it out. We have revised it (Figure 5B).
Figure 5A and 5B probably come from the same experiment. The amount of sample in Figure 5A and Figure 5B are inconsistent
Response: Thanks for point it out. We have revised it (Figures 5A and 5B).
Round 2
Reviewer 2 Report
Comments and Suggestions for Authors
The authors have responded with comments and revised the manuscript. I recommend acceptance.
Author Response
Response to Reviewers:
Reviewer #2: The authors have responded with comments and revised the manuscript. I recommend acceptance.
Response: We thank the reviewer #2 for his/her great comments. We have revised the manuscript accordingly.